# Regional differences in hamstring muscle damage after a marathon

**Ayako Higashihara**[1]*, **Kento Nakagawa**[2], **Takayuki Inami**[1], **Mako Fukano**[3], **Satoshi Iizuka**[2], **Toshihiro Maemichi**[2], **Satoru Hashizume**[4], **Takaya Narita**[5], **Norikazu Hirose**[2]

**1** Institute of Physical Education, Keio University, Kanagawa, Japan, **2** Faculty of Sport Sciences, Waseda University, Saitama, Japan, **3** Shibaura Institute of Technology, Tokyo, Japan, **4** National Institute of Advanced Industrial Science and Technology, Tokyo, Japan, **5** Department of Sport Technology, Toin University of Yokohama, Kanagawa, Japan

* higashihara@keio.jp

**Data Availability Statement:** All relevant data are uploaded to Zenodo (DOI: 10.5281/zenodo. 3734098).

**Funding:** This work was supported by the Mizuno Sports Promotion Foundation.

## Abstract

Previous studies suggest that marathon running induces lower extremity muscle damage. This study aimed to examine inter- and intramuscular differences in hamstring muscle damage after a marathon using transverse relaxation time ($T_2$)–weighted magnetic resonance images (MRI). 20 healthy collegiate marathon runners (15 males) were recruited for this study. $T_2$-MRI was performed before (PRE) and at 1 (D1), 3 (D3), and 8 days (D8) after marathon, and the $T_2$ values of each hamstring muscle at the distal, middle, and proximal sites were calculated. Results indicated that no significant intermuscular differences in $T_2$ changes were observed and that, regardless of muscle, the $T_2$ values of the distal and middle sites increased significantly at D1 and D3 and recovered at D8, although those values of the proximal site remained constant. $T_2$ significantly increased at distal and middle sites of the biceps femoris long head on D1 ($p = 0.030$ and $p = 0.004$, respectively) and D3 ($p = 0.007$ and $p = 0.041$, respectively), distal biceps femoris short head on D1 ($p = 0.036$), distal semitendinosus on D1 ($p = 0.047$) and D3 ($p = 0.010$), middle semitendinosus on D1 ($p = 0.005$), and distal and middle sites of the semimembranosus on D1 ($p = 0.008$ and $p = 0.040$, respectively) and D3 ($p = 0.002$ and $p = 0.018$, respectively). These results suggest that the distal and middle sites of the hamstring muscles are more susceptible to damage induced by running a full marathon. Conditioning that focuses on the distal and middle sites of the hamstring muscles may be more useful in improving recovery strategies after prolonged running.

## Introduction

Prolonged running, such as in a marathon, has become a popular sports activity for health promotion. However, marathon races are often reported to induce muscle damage in the lower extremity muscles, manifesting as decreased muscle strength [1–4], occurrence of delayed onset muscle soreness [4, 5], and increased plasma creatine kinase (CK) levels [1, 2, 4], which

**Competing interests:** The authors have declared that no competing interests exist.

usually peak at 1–3 days after the race and take several days to recover. To better understand the location of muscle damage in the leg muscles induced by a marathon, previous studies using ultrasound elastography have demonstrated that the thigh and lower leg muscles became harder after a marathon [6, 7], that the amount of change in muscle mechanical properties (e.g., muscle hardness) does not occur uniformly within the lower extremity muscles, and that the recovery time for each muscle varies. Muscle hardness in the vastus lateralis, biceps femoris, and soleus muscles returned to baseline at 8 days after the race, whereas the rectus femoris and gastrocnemius medial muscle hardness did not recover even after 8 days [6]. A study which determines each muscle damage after prolonged running can provide evidence for recreational and competitive runners about the time required for the recovery of each lower extremity muscle between training sessions and/or races.

Muscle functional magnetic resonance imaging (mfMRI) allows for the examination of differences in the intensity and/or pattern of exercise-induced muscle damage. This method relies on an exercise-induced increase in the proton transverse relaxation time ($T_2$) of the muscle on MRI, which could provide information about the water content of muscle tissues. A delayed $T_2$ increase, which occurs at 1–5 days after exercise, is a consequence of muscle damage (inflammatory oedema) induced by eccentric contractions [8]; a high correlation has been reported between changes in $T_2$ and plasma CK levels measured within several days after eccentric exercise [9]. In addition, since $T_2$ values are mapped out across cross-sectional images of muscles, mfMRI can determine muscle damage even in deep muscles; this high spatial resolution overcomes the limitations associated with ultrasound imaging. By using these advantages, a previous study examined the location of muscle damage within the quadriceps femoris muscles induced by downhill running and suggested that muscle damage is specifically located at the proximal and middle sites of the vastus intermedius muscle compared to other knee extensors [10]. Considering these findings, it may be possible that different patterns of running-induced muscle damage may also exist in each section of the hamstring muscles.

The hamstring muscles are composed of three muscles: the semimembranosus (SM), semitendinosus (ST), and biceps femoris (BF); the BF has a short (BFsh) and a long head (BFlh). The BFsh arises from the femur and shares a common distal tendon with BFlh, making it a monoarticular muscle that spans the knee joint only. By contrast, the SM, ST, and BFlh all arise from the ischial tuberosity on the pelvis and thus are biarticular muscles that span the hip and knee joints. During running, the hamstring muscles play an important role in decelerating the flexing hip and rapidly extending the knee [11]; biarticular hamstring muscles contract eccentrically in the terminal swing phase of running [12, 13]. In addition, hamstring muscles are known to be susceptible to muscle strain. Repetitive eccentric contractions associated with running may lead to the accumulation of eccentrically induced muscle damage [14, 15]. Such accumulation could, in turn, leave the hamstring muscles more at risk of strain injury [14]. Considering the fact that hamstring muscles are susceptible to muscle strain, identification of the sites in the hamstring muscles that are significantly damaged by prolonged running would contribute to an effective strategy for recovery focusing on injury prevention. In addition, this will provide evidence about the time required for the recovery of each site in the hamstring muscle between training sessions and/or competitions. Nevertheless, to our knowledge, there have been no studies on the location of muscle damage in the hamstring muscles induced by repetitive eccentric contractions after marathon running.

Thus, this study aimed to identify the sites in the hamstring muscles that are significantly damaged by prolonged running (full marathon) using mfMRI. For this purpose, we measured running-induced changes in $T_2$ values at the proximal, middle, and distal sites of each hamstring muscle. We hypothesized that the location of muscle damage following running would be greatest in the BFlh muscle, based on previous kinematic analyses of running, considering

that the peak muscle–tendon stretch is greatest in the BFlh muscle among the biarticular hamstring muscles during the terminal swing phase of running [12, 13], which occurs simultaneously with high muscle activity [16]. In addition, given that the knee joint performs greater negative (eccentric) work than the hip joint during running [17], we hypothesized that muscle damage would be more pronounced in the distal site of the biarticular hamstring muscles.

## Materials and methods

### Subjects

Twenty collegiate runners (males, 15; females, 5) without lower extremity injuries were recruited for this study. Mean age, height, and body mass of participants were 20.5±1.4 years, 168.6±7.1 cm, and 59.4±7.1 kg, respectively. Participants were members of a track and field club who had registered for a marathon race (Fujisan Marathon in Japan, 2016–2017; 42.195 km) and had no history of lower limb injuries.

This study was approved by the local human research ethics committee of Waseda University (2014–246), complied with their requirements for human experimentation, and conformed to the principles of the Declaration of Helsinki. The purpose, procedures, and risks of the study were communicated to the participants, and written informed consent was obtained from each participant. Participants were asked not to perform exercise, such as running, and not to receive special care for recovery, such as a massage, during the observation period after marathon.

### Procedures

**Data collection.** All participants completed the marathon. Maximal isometric knee flexion torque and $T_2$-weighted magnetic resonance (MR) images of the thigh were obtained before (2 days before the race [PRE]) and at 1 (D1), 3 (D3), and 8 (D8) days after the marathon race. The course altitude was approximately 850 m above sea level from the start to approximately 22 km, increased to around 910 m, was almost flat from 23 km to 34 km, declined to 850 m from 34 km to 36 km, and was flat from 36 km to finish.

**$T_2$-MRI.** Thigh $T_2$-MRI (Fig 1) was obtained with an MR scanner (Signa EXCITE 1.5T; GE Medical Systems, Waukesha, WI). Before scanning, participants lay quietly for 10 min. Ink lines were drawn transversely across the middle (50%) of the thigh (i.e., the distance from the great trochanter of the femur to the articular cleft between the femoral and tibial condyles) and oil capsules were put as markers on the skin surface at the lateral side. Participants lay supine with their legs fully extended and muscles relaxed in a magnet bore. The images were acquired with the following parameters: echo times, 25, 50, 75, and 100 ms; repetition time, 2000 ms; matrix, 256×160; field of view, 240 mm; slice thickness, 10 mm; gap, 10 mm. Images were analysed with ImageJ software (National Institute of Health, Bethesda, MD). Regions of interest were drawn in each slice by manually tracing the border of the anatomical cross-sectional area of each of the hamstring muscles at the middle (50%), proximal (8 cm proximal from the middle site), and distal (8 cm distal from the middle site) thigh sites (Fig 1). Care was taken to exclude noncontractile tissues, such as intramuscular fat and blood vessels. The $T_2$ values for each pixel within the BFlh, BFsh, ST, and SM muscles were calculated, and the mean value was computed for each slice. As the BFsh is a monoarticular muscle with an origin at a more distal site than those of the other three muscles, only the middle and distal sites were analysed. $T_2$ relaxation time was calculated by least-squares analysis, fitting the signal intensity at each of the four echo times (n×25 ms: 25, 50, 75, and 100 ms) to a monoexponential decay

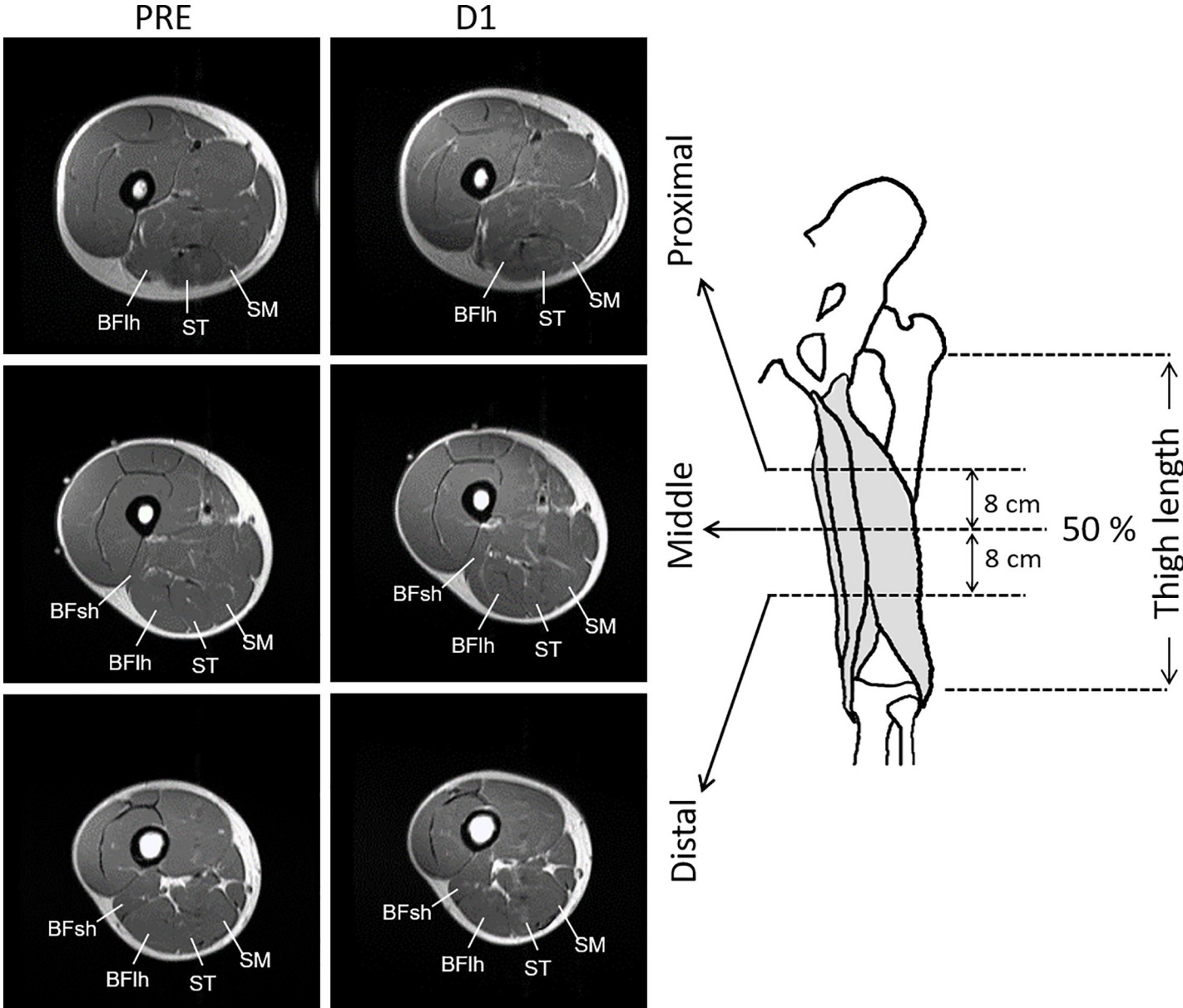

**Fig 1. MRI measurement sites. Examples of T$_2$-MRI at the proximal, middle, and distal sites scanned before (PRE) and at 1 day (D1) after a full marathon race.**
BFlh, biceps femoris long head; BFsh, biceps femoris short head; MRI, magnetic resonance imaging; SM, semimembranosus; ST, semitendinosus.

using the following equation:

$$S_n = S_0 \exp^{(-TE_n/T_2)} \ (n = 25, \ 50, \ 75, \text{ and } 100 \text{ ms}),$$

where TE is the echo time, $S_0$ is the signal intensity at 0 ms, and $S_n$ is the signal intensity at $TE_n$.

The aforementioned analyses were performed three times for each slice and the average value was used for further analysis. The coefficient of variation of the three measurements for T$_2$ values was 0.8±0.8%. The intraclass correlation coefficient of the measurements was 0.964. The absolute T$_2$ values (ms) were used in between-time comparisons for each site. In addition,

as the timing of $T_2$ manifestations could vary among sites and individuals, peak $T_2$ values (peak $\Delta T_2$, in ms) were calculated by subtracting the baseline (PRE) value from the peak value after marathon for each site, irrespective of the time point at which peak value occurred, and was used for between-site comparisons.

**Maximal isometric knee flexion torque.** After MR scanning, participants performed isometric right knee flexion with maximal effort while lying down in a prone position. The experimenter fixed the participant's knee joint at 90° with a hand-held dynamometer (microFET2; Hoggan Scientific, Salt Lake, UT); the participants performed two isometric contractions against the experimenter's force. They were asked to develop torque gradually over 5 s and reach the maximum, and they received verbal encouragement to sustain the maximum effort for 2 s from the examiner. Two peak torque values were averaged for the statistical analysis.

## Statistical analyses

All statistical analyses were conducted using SPSS version 14.0 (IBM Corp., Armonk, NY). Changes in $T_2$ values were compared by a three-way analysis of variance (ANOVA) with repeated measures (muscle × site × time point). When significant interaction effects were found, Bonferroni's post hoc test was performed to compare changes from the PRE for each site. Absolute peak $\Delta T_2$ values were compared among the sites using one-way ANOVA (11 sites) followed by Bonferroni post hoc testing. Changes in maximal knee flxexion torque were compared by a one-way (4 time points) repeated measures ANOVA followed by Bonferroni post hoc testing. Partial $\eta^2$ (for ANOVA) was calculated as indices of effect size (ES) for the ANOVA. To report the ES for post hoc comparisons, the mean change value from the PRE divided by the standard deviation of the change value was calculated [18]. Statistical significance was set at $p < 0.05$.

## Results

### Marathon time

The average marathon completion time was 4 h, 7 min, 55 s ± 47 min, 39 s.

**$T_2$-MRI.** Three-way ANOVA analysis for $T_2$ values indicated some interaction effects: muscle × site (F = 3.393, $p$ = 0.006) and site × time point (F = 5.729, $p < 0.001$) (Table 1). The absolute $T_2$ values for each site are shown in the Table 2. Fig 2 shows the mean difference and 95% confidence intervals in $T_2$ value from the baseline (PRE) for each muscle. Bonferroni's post hoc analysis indicated that, regardless of muscle, the $T_2$ values of the distal and middle sites increased significantly at D1 and D3 and recovered at D8 (no significant difference on D8 when compared with PRE), although those values of the proximal site remained constant. $T_2$ significantly increased at distal and middle sites of the biceps femoris long head on D1 ($p$ = 0.030 and $p$ = 0.004, respectively) and D3 ($p$ = 0.007 and $p$ = 0.041, respectively), distal biceps femoris short head on D1 ($p$ = 0.036), distal semitendinosus on D1 ($p$ = 0.047) and D3 ($p$ = 0.010), middle semitendinosus on D1 ($p$ = 0.005), and distal and middle sites of the semimembranosus on D1 ($p$ = 0.008 and $p$ = 0.040, respectively) and D3 ($p$ = 0.002 and $p$ = 0.018, respectively). There was no significant difference in the $T_2$ value of the middle BFsh muscle and the proximal site of the BFlh, ST, and SM at any time point compared with that of PRE. Fig 3 shows the peak $\Delta T_2$ value for each site. No significant differences in the peak $\Delta T_2$ values after the full marathon were observed among sites.

**Maximal isometric knee flexion torque.** One-way ANOVA revealed a significant main effect of time on maximal knee flexion torque (F = 8.491, $p < 0.001$, partial $\eta^2$ = 0.309). Bonferroni post-hoc testing showed that the knee flexion torque significantly decreased compared with the baseline (PRE) value on D1 (−22.7%, $p$ = 0.007, $ES$ = 0.85) and D3 (−17.7%, $p$ = 0.009,

**Table 1. Results of statistical analyses for the $T_2$ values.**

| Source | Three-way ANOVA | | | | |
|---|---|---|---|---|---|
| | *df* | **F-value** | *p*-value | $\eta^2$ | |
| Site | 2 | 14.950 | 0.000 | 0.125 | ** |
| Muscle | 3 | 21.158 | 0.000 | 0.233 | ** |
| Time point | 3 | 27.508 | 0.000 | 0.116 | ** |
| Site × muscle | 5 | 3.393 | 0.006 | 0.075 | * |
| Site × time point | 6 | 5.729 | 0.000 | 0.052 | ** |
| Muscle × time point | 9 | 0.368 | 0.950 | 0.005 | n.s. |
| Site × muscle × time point | 15 | 0.861 | 0.609 | 0.020 | n.s. |

* $p < 0.01$

** $p < 0.001$; n.s. not significantly different

*ES* = 0.83) after marathon; however, there was no significant difference on D8 (−5.5%, $p$ = 0.050, *ES* = 0.65) compared with PRE (Fig 4).

## Discussion

We determined whether there are inter- and intramuscular differences in damage to the hamstring muscles after a full marathon using $T_2$-MRI. The main findings of the present study were that no significant intermuscular differences in the magnitude of $T_2$ change after a marathon were found and that $T_2$ significantly increased at the distal and middle sites of the biarticular hamstring muscles on D1 and D3 after the marathon race; however, no significant difference in the $T_2$ value of the proximal site was observed at any time point when compared with PRE. These results suggest that running a full marathon induces muscle damage

**Table 2. Time-dependent change in the absolute value of $T_2$ values (Mean ± SD) measured before (PRE) and 1 day (D1), 3 days (D3) and 8 days (D8) after the marathon race.** Effect sizes are shown in parentheses.

| | | PRE | D1 | D3 | D8 |
|---|---|---|---|---|---|
| BFlh | | | | | |
| | Distal | 31.7 ± 1.2 | 33.6 ± 2.7* [0.71] | 33.5 ± 2.0** [0.86] | 32.1 ± 1.9 [0.22] |
| | Middle | 32.2 ± 1.1 | 33.4 ± 1.5** [0.90] | 33.1 ±1.4* [0.68] | 32.8 ± 2.3 [0.29] |
| | Proximal | 35.5 ±1.8 | 36.3 ±2.1 [0.34] | 35.7 ± 1.7 [0.13] | 34.8 ± 1.3 [0.41] |
| BFsh | | | | | |
| | Distal | 32.2 ± 1.6 | 33.9 ± 2.9* [0.69] | 34.1 ± 3.2 [0.22] | 33.5 ± 3.4 [0.01] |
| | Middle | 32.8 ± 1.9 | 34.0 ± 2.5 [0.54] | 33.2 ± 2.3 [0.56] | 32.8 ± 2.2 [0.38] |
| ST | | | | | |
| | Distal | 30.4 ± 1.6 | 32.3 ± 2.7* [0.66] | 32.1 ± 2.0* [0.82] | 31.5 ± 2.4 [0.47] |
| | Middle | 30.9 ± 1.1 | 32.0 ± 1.5** [0.89] | 31.6 ± 1.7 [0.58] | 31.6 ± 2.0 [0.43] |
| | Proximal | 32.2 ± 2.0 | 32.5 ± 1.4 [0.3] | 31.3 ± 1.5 [0.53] | 31.0 ±1.2 [0.51] |
| SM | | | | | |
| | Distal | 31.5 ± 1.5 | 33.2 ± 2.5** [0.84] | 33.1 ± 1.6** [1.00] | 32.2 ± 2.2 [0.33] |
| | Middle | 31.7 ± 1.1 | 32.7 ± 1.4* [0.68] | 32.7 ± 1.3* [0.76] | 32.3 ± 1.9 [0.26] |
| | Proximal | 33.8 ± 2.9 | 34.1 ± 3.1 [0.15] | 33.5 ± 3.2 [0.11] | 33.3 ± 2.2 [0.24] |

*Significantly different from PRE at $p < 0.05$.

**Significantly different from PRE at $p < 0.01$. BFlh, biceps femoris long head; BFsh, biceps femoris short head; ST, semitendinosus; SM, semimembranosus.

(A) BFlh

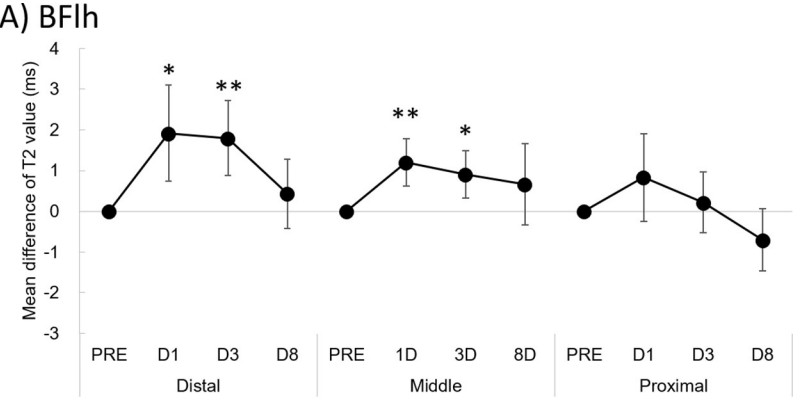

(B) BFsh

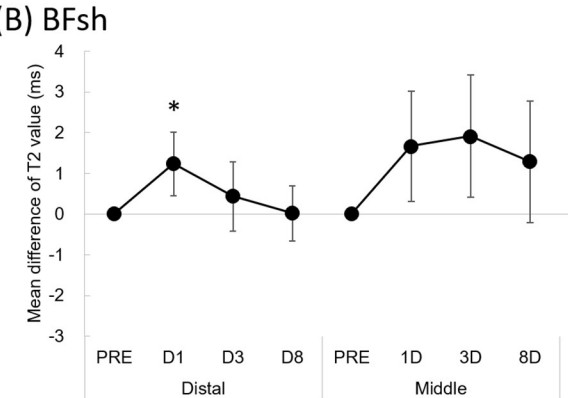

(C) ST

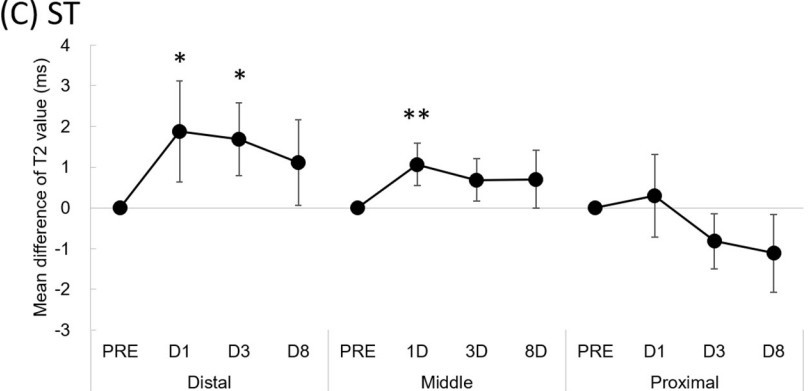

(D) SM

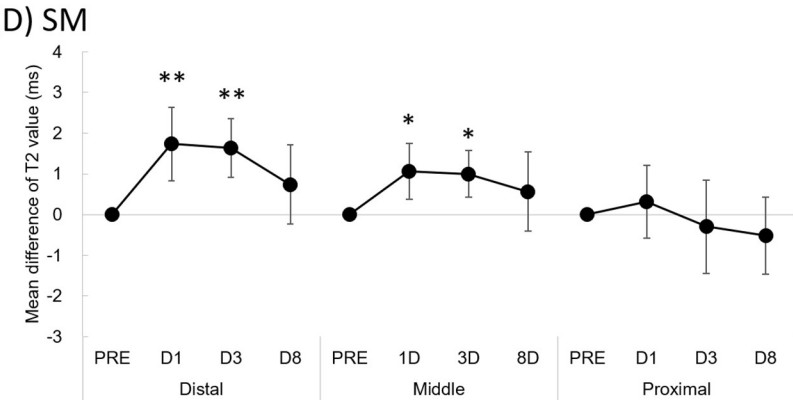

**Fig 2.** Mean difference in $T_2$ value from the baseline value (PRE) of the biceps femoris long head (A), biceps femoris short head (B), semitendinosus (C), and semimembranosus (D) muscles. Error bars indicate 95% confidence intervals. *Significantly different from PRE at $p < 0.05$. **Significantly different from PRE at $p < 0.01$. D1, 1 day after marathon; D3, 3 days after marathon; D8, 8 days after marathon. BFlh, biceps femoris long head; BFsh, biceps femoris short head; ST, semitendinosus; SM, semimembranosus.

(inflammatory oedema) appearing as a $T_2$ increase in the hamstring muscles and that the magnitude of muscle damage induced by prolonged running would be greater in the distal and middle sites than in the proximal site of the hamstring muscles.

Marathon running resulted in significant decreases in maximal knee flexion torque on D1 and D3 (Fig 4). This result is similar to that of a previous study, which showed a significant reduction in maximal voluntary knee flexion torque following a full marathon [4]. However, no significant differences in the magnitude of $T_2$ change (peak $\Delta T_2$) after running were found among muscles (Fig 3). These results did not support our hypothesis that muscle damage would be greatest in the BFlh muscle. We drafted this hypothesis based on previous observations, which suggested that the magnitude of muscle–tendon strain during contraction is the more relevant parameter for muscle damage [19] and that peak muscle–tendon stretch is greatest in the BFlh during running compared to the ST and SM [12, 13]. Schache et al. [12] investigated the peak muscle–tendon stretch of the biarticular hamstring muscles at different running speeds and demonstrated that the peak stretch was greater for BFlh compared to ST and SM across a range of running speeds. They also found that there was no statistical difference in the magnitude of the maximum stretch between these muscles during slower speed below 18 km/h. The participants in the present study completed a full marathon at an average time of 4 h, 7 min, 55 s ± 47 min, 39 s (i.e., at an average speed of 11 km/h). Thus, the findings of Schache et al. [12] would support the result of the current study; however, the degree of muscle damage induced by prolonged running, such as after a full marathon, may not simply

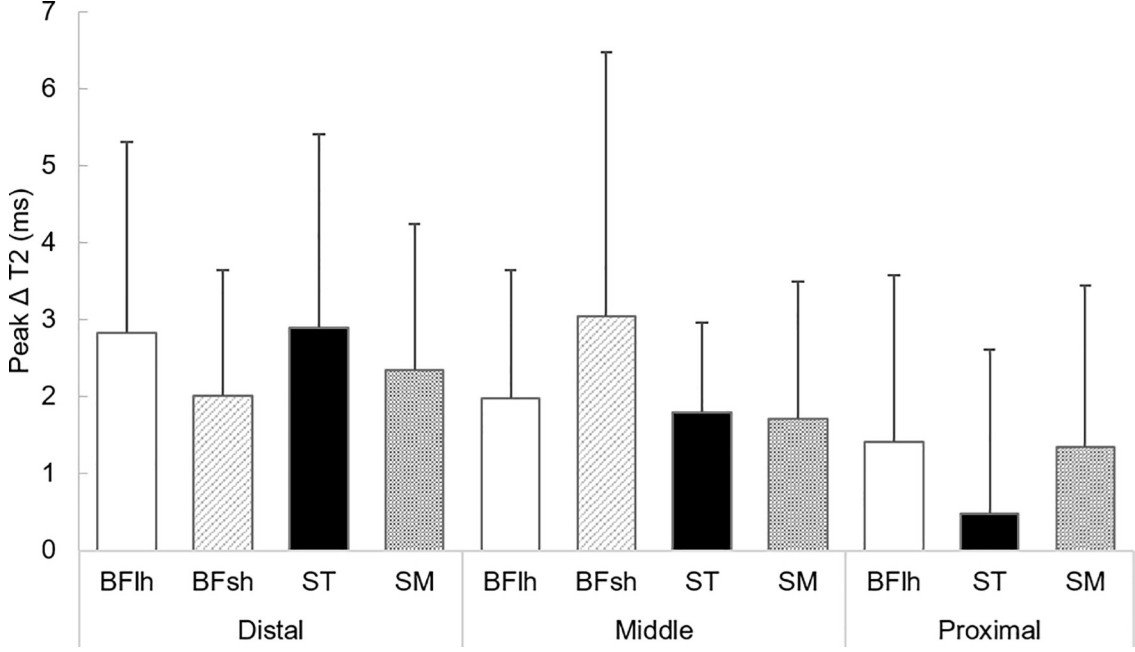

**Fig 3. Peak $T_2$ changes from PRE ($\Delta T_2$) within 8 days after a full marathon.** Values are mean±standard deviation. BFlh, biceps femoris long head; BFsh, biceps femoris short head; ST, semitendinosus; SM, semimembranosus.

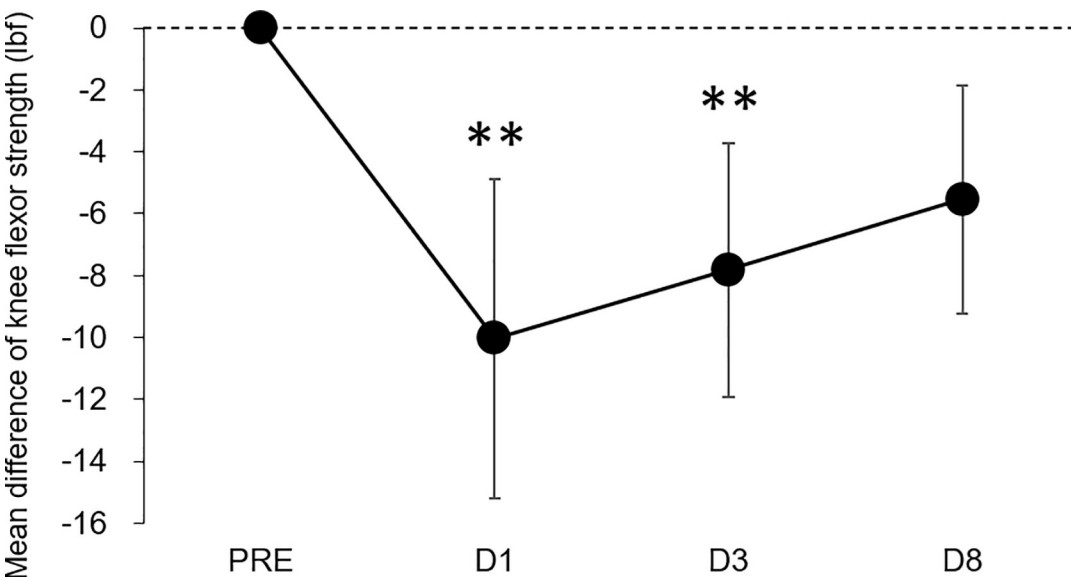

**Fig 4. Mean difference in maximal isometric knee flexion torque from the baseline (PRE).** Error bars indicate 95% confidence intervals. D1, 1 day after marathon; D3, 3 days after marathon; D8, 8 days after marathon.

be explained based on the intermuscular difference in stretching magnitude: one possibility is the alternate muscle activity among the synergists due to the fatigue during prolonged running, while the previous study suggested the alterations in the neuromuscular activation patterns among the quadriceps muscles during the repetitive fatiguing exercise [20]. This phenomenon may occur in the hamstring muscles during marathon running. However, given the results of the mfMRI analysis indicating the intensity and/or pattern of exercise-induced muscle damage after exercise, whether the alternate muscle activity among the hamstring muscles occurs during the marathon race is uncertain. Even though it is a speculation, is worthy of further examination.

$T_2$ values significantly increased at the distal and middle sites of the BFlh, ST, and SM muscles on D1 and D3 after running (Table 2, Fig 2), with a pronounced effect size found, whereas the $T_2$ value of the proximal site at any time point did not significantly increase compared with PRE. This suggests that the distal and middle sites of the biarticular hamstring muscles are more susceptible to damage after a marathon. The amount of negative work performed by the muscle would be the best predictor of the magnitude of damage, as indicated by force deficits following active or passive muscle stretching [21]. Therefore, a possible explanation for the present results is that the location of muscle damage after prolonged running is primarily influenced by the intensity of repetitive negative work performed by the lower extremity. During running, the knee joint performs greater negative work than the hip joint (e.g., −0.41 J/kg for the knee and −0.10 J/kg for the hip at a running speed of 3.5 m/s) [17]. In addition, a previous study reported that during eccentric knee flexion, the greatest tissue motion in the BFlh muscle is observed along the distal musculotendinous junction [22]. Considering these findings and the fact that a greater degree of muscle tissue strain during eccentric contraction causes muscle damage [19], the magnitude of muscle damage induced by prolonged running would be greater in the distal than in the proximal portions of the hamstring muscles.

A previous study suggested that the accumulation of eccentrically-induced muscle damage associated with running may leave the hamstring muscles more vulnerable to strain injury [14]. The aetiology and risk of hamstring strain injury is closely related to high-speed running

[23], and transiently elevated high-speed running exposure (<24 km/h) at 7–14 days prior to injury increases the likelihood of injury [24, 25]. However, sports in which a high rate of hamstring strain injury has been reported [26], such as Australian football, require prolonged and high-intensity intermittent performance, and players cover long distances during a game (12,939±1145 m for total distance) at various speeds; movement distance with a higher-speed running (>14.5 km/h) accounts only for 30% of the total distance [27]. The present results indicated that repetitive eccentric contractions associated with lower speed (at an average speed of 11 km/h) but prolonged running, such as in a marathon, produce inflammatory oedema at the distal and middle sites of biarticular hamstring muscles. In addition, considering that this microscopic damage in hamstring muscles takes several days to recover, accumulation of prior running exposure even at lower speeds may contribute to the susceptibility of strain injury for some time after. This hypothesis is yet to be explored. Moreover, the magnitude of muscle damage is largely reduced when the same or similar eccentric exercise is repeated within several weeks [28]. Therefore, differences in training and competition level may affect the magnitude of muscle damage induced by prolonged running. This should be regarded as a limitation; future research should ascertain whether similar findings are also seen in runners from different competitive levels or other competitive athletes.

In this study, we demonstrate that the distal and middle sites of the biarticular hamstring muscles are more susceptible to damage induced after running a marathon. Based on our results, conditioning that focuses on the distal and middle sites of the hamstring muscles may be more useful in improving recovery strategies as well as for preventing injury after prolonged running. In addition, our results showed that inflammatory oedema appearing as $T_2$ increase takes several days to recover. Therefore, accumulation of low-intensity running sessions may contribute to the susceptibility of strain injury afterwards. However, our results are inconsistent with the clinical observations that the proximal muscle–tendon junction of the BFlh muscles is more commonly affected in acute hamstring muscle injury [23, 29]. This discrepancy could be explained by the possible difference in the neuromechanical behaviour and joint kinetics between high-speed running, in which an injury occurs, and prolonged running. Therefore, further research is required to identify the regional damage patterns after repetitive high-speed running relative to the injury site. It is likely that such studies can provide further insights into the mechanisms of muscle strain injury during high-speed running.

In conclusion, the results indicated that marathon running induces a significant damage to hamstring muscles; however, the amount of change in inflammatory oedema appearing as $T_2$ increase did not differ among hamstring muscles. Furthermore, site-dependent changes in $T_2$ values after a full marathon were observed within muscles; that is, pronounced inflammatory oedema was observed in the distal and middle sites of the biarticular hamstring muscles. Our findings provide a better understanding of the site-specific muscle damage patterns of the hamstring muscle induced by prolonged running.

## Author Contributions

**Conceptualization:** Ayako Higashihara, Kento Nakagawa, Takayuki Inami.

**Data curation:** Ayako Higashihara.

**Formal analysis:** Ayako Higashihara.

**Funding acquisition:** Kento Nakagawa.

**Investigation:** Ayako Higashihara, Kento Nakagawa, Takayuki Inami, Mako Fukano, Satoshi Iizuka, Toshihiro Maemichi, Satoru Hashizume.

**Methodology:** Ayako Higashihara.

**Project administration:** Takaya Narita, Norikazu Hirose.

**Resources:** Ayako Higashihara, Takaya Narita.

**Validation:** Ayako Higashihara, Kento Nakagawa.

**Visualization:** Ayako Higashihara.

**Writing – original draft:** Ayako Higashihara.

**Writing – review & editing:** Ayako Higashihara, Kento Nakagawa, Takayuki Inami, Mako Fukano, Satoshi Iizuka, Toshihiro Maemichi, Takaya Narita, Norikazu Hirose.

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
