## [Decision Letter · Decision Letter 0]

27 Mar 2020

PONE-D-20-03671

Regional differences in hamstring muscle damage after a marathon

PLOS ONE

Dear Ms. Higashihara,

Thank you for submitting your manuscript to PLOS ONE.  I sincerely apologize for the delay in reviewing your manuscript. After careful consideration, we feel that it has merit but does not fully meet PLOS ONE’s publication criteria as it currently stands. Therefore, we invite you to submit a revised version of the manuscript that addresses the points raised during the review process.

Please carefully consider the following points along with that of the other reviewers prior to resubmitting: 

1) Per PLOS ONE policy, the authors  must make the data available to the extend that the analysis can be reproduced. Currently, this has not been done. 

2) The analysis does not appear to match your study purpose. Please alter this prior to resubmitting. The document states that the study aimed to identify sites (proximal, distal, middle) in the hamstring muscles (biceps femoris, semitendinosus, etc) in the days following a marathon (Pre, d1, d3, d8). It appears that the purpose was to determine differences between all of these but the current model does not appear to account for that. It seems that you have a Muscle x Site x Time design. This, I think, would better address your research question. 

3) please explicitly state how Cohen's d was calculated.   

4) It might be helpful for you to represent Figure 2 as change scores (variability of the change in 95% CI). This would seemingly better highlight the variability of interest. For example, the variability of the sample at Pre is not relevant to the research question. What is relevant, is the variability of the change at that site following the marathon. 

We would appreciate receiving your revised manuscript by May 11 2020 11:59PM. To enhance the reproducibility of your results, we recommend that if applicable you deposit your laboratory protocols in protocols.io, where a protocol can be assigned its own identifier (DOI) such that it can be cited independently in the future. For instructions see: http://journals.plos.org/plosone/s/submission-guidelines#loc-laboratory-protocols

We look forward to receiving your revised manuscript.

Kind regards,

Jeremy P Loenneke

Academic Editor

PLOS ONE

Journal Requirements:

2. Thank you for submitting the above manuscript to PLOS ONE. During our internal evaluation of the manuscript, we found significant text overlap between your submission and the following previously published works:

https://link.springer.com/article/10.1007%2Fs00421-013-2713-9

https://onlinelibrary.wiley.com/doi/full/10.1111/sms.12880

Please revise the manuscript to rephrase the duplicated text, cite your sources, and provide details as to how the current manuscript advances on previous work. Please note that further consideration is dependent on the submission of a manuscript that addresses these concerns about the overlap in text with published work.

Reviewers' comments:

Reviewer's Responses to Questions

**Comments to the Author**

1. Is the manuscript technically sound, and do the data support the conclusions?

Reviewer #1: Yes

Reviewer #2: Yes

2. Has the statistical analysis been performed appropriately and rigorously? 

Reviewer #1: Yes

Reviewer #2: Yes

3. Have the authors made all data underlying the findings in their manuscript fully available?

Reviewer #1: No

Reviewer #2: Yes

4. Is the manuscript presented in an intelligible fashion and written in standard English?

Reviewer #1: Yes

Reviewer #2: Yes

5. Review Comments to the Author

Reviewer #1: This study used mfMRI to examine muscle damage in sites along the hamstring muscles following marathon running. The “experiments, statistics, and other analyses” appear to have been “performed to a high technical standard and are described in sufficient detail” as required by PLOS ONE. The conclusions are logical and the article is well-written. This study provides novel and useful original research. This reviewer sees no reason for this article to not be published as is.

Reviewer #2: Overall, the manuscript is well written. The study was nicely designed and the results were also convincing. This reviewer has only a couple of minor questions that need to be clarified.

1. Why muscle soreness was not measured given the fact that muscle soreness is also a commonly used indirect marker for exercise-induced muscle damage?

2. Line 50:

“recovery of the rectus femoris and gastrocnemius medial muscle hardness was not observed “

Does that mean the rectus femoris and gastrocnemius medial muscle got permanent muscle damage? If not, what does the sentence mean?

3. Line 106-107: Participants were asked not to perform exercise, such as running…”

How long (e.g. # of days prior and after the marathon running) were the participants not allowed to perform exercise? Were these participants trained or untrained people? If they were trained people, did the exercise prohibition of from this study induce any detraining or other effect(s)?

4. Line 113: The baseline measurements were obtained 2 days before the race. Why 2 days prior specifically? Why not 1 day or any other time before the race?

5. Line 160 and 162: Why two different post-hoc testings (Bonferroni and Dunnett)? Why not use only one of them and keep it consistent?

6. Lines 218-226. The discussion of the non-significant intermuscular damage after marathon running needs to be expanded and more thorough. If this cannot be explained only by stretching magnitude, what else could be the mechanism/explanation?

6. PLOS authors have the option to publish the peer review history of their article (what does this mean?). If published, this will include your full peer review and any attached files.

Reviewer #1: No

Reviewer #2: No

---

## [Author Response · Author response to Decision Letter 0]

17 Apr 2020

Responses to Editor: 

We appreciate your insightful comments that helped improve the quality of our manuscript. We have revised the manuscript in accordance with your suggestions and hope that the revised manuscript is eligible for publication.

Our answers to the comments raised in the editorial report are as follows.

1) Per PLOS ONE policy, the authors must make the data available to the extend that the analysis can be reproduced. Currently, this has not been done. 

We have uploaded all relevant data to Zenodo at https://zenodo.org/record/3659388#.XoL9Roj7RPY.

2) The analysis does not appear to match your study purpose. Please alter this prior to resubmitting. The document states that the study aimed to identify sites (proximal, distal, middle) in the hamstring muscles (biceps femoris, semitendinosus, etc) in the days following a marathon (Pre, d1, d3, d8). It appears that the purpose was to determine differences between all of these but the current model does not appear to account for that. It seems that you have a Muscle x Site x Time design. This, I think, would better address your research question. 

Thank you for your insightful comments regarding data analysis. We have conducted three-way ANOVA analysis (muscle × site × time point) for the T2 values and added Table 1 to display the results of statistical analysis in the revised manuscript. Three-way ANOVA found some interaction effects: muscle × site (F = 3.393, p = 0.006) and site × time point (F = 5.729, p < 0.001). Post hoc analysis indicated that, regardless of muscle, the T2 values of the distal sites increased significantly at D1 and D3, although those values of the medial and proximal sites remained constant. Regarding the intermuscular differences, the results of the peak T2 changes from PRE (ΔT2) shows no significant intermuscular differences in T2 changes. This would support the result that intermuscular differences in magnitude of the hamstring muscle damage was not observed after marathon race.

The results were slightly altered; however, the directions of the discussion and conclusion are almost the same to the previous version.

3) please explicitly state how Cohen's d was calculated.

We have added the formula that was used for the calculation of the Cohen’s d values as follow:

[Lines 163-169] To report the ES for post hoc comparisons, Cohen’s d was calculated (Eq. (1)) and interpreted as large (≥ 0.80), medium (0.50–0.79), small (0.20–0.49), and trivial (<0.20) [18]:

d= (y ®_B (q)-y ®_A (q))/√((s_A^2 (q)+s_B^2 (q))/2) (1)

where, y ®_A (q) and y ®_b (q) are the pointwise means for the values of the baseline and each time point, and s_A (q) and s_B (q) are the pointwise standard deviations for the baseline and each time point, respectively.

4) It might be helpful for you to represent Figure 2 as change scores (variability of the change in 95% CI). This would seemingly better highlight the variability of interest. For example, the variability of the sample at Pre is not relevant to the research question. What is relevant, is the variability of the change at that site following the marathon. 

Thank you for your insightful suggestion regarding the figure. We have revised the Figure 2 in accordance with your suggestions. We also revised the Figure 4 showing maximal isometric knee flexion torque as well. Error bars indicate 95% confidence intervals in Figures 2 and 4.

Revised Figure2 Revised Figure 4

Responses to Reviewer #1: 

Reviewer #1: This study used mfMRI to examine muscle damage in sites along the hamstring muscles following marathon running. The “experiments, statistics, and other analyses” appear to have been “performed to a high technical standard and are described in sufficient detail” as required by PLOS ONE. The conclusions are logical and the article is well-written. This study provides novel and useful original research. This reviewer sees no reason for this article to not be published as is.

We appreciate your comments. We hope our findings provide more useful information in improving recovery strategies after prolonged running.

Responses to Reviewer #2: 

Reviewer #2: Overall, the manuscript is well written. The study was nicely designed and the results were also convincing. This reviewer has only a couple of minor questions that need to be clarified.

We appreciate your insightful comments that have helped improve the quality of our manuscript. We have revised the manuscript in accordance with your suggestions and hope that the revised manuscript is eligible for publication.

Our answers to the comments raised in the editorial report are as follows:

1. Why muscle soreness was not measured given the fact that muscle soreness is also a commonly used indirect marker for exercise-induced muscle damage?

This study aimed to examine inter- and intramuscular differences in hamstring muscle damage after a marathon using mfMRI. Since mfMRI can determine muscle damage even in deep muscles (e.g., biceps femoris short head in this study), this high spatial resolution overcomes the limitations associated with the muscle soreness. Therefore, the present study examined the location of muscle damage within the hamstring muscles by using mfMRI.

　

2. Line 50:

“recovery of the rectus femoris and gastrocnemius medial muscle hardness was not observed “

Does that mean the rectus femoris and gastrocnemius medial muscle got permanent muscle damage? If not, what does the sentence mean?

Your understanding is correct. The rectus femoris and gastrocnemius medial muscle hardness significantly increased in 1 day and 3 days after the marathon race and did not recover to the baseline value even after 8 days. We have revised the sentence in the manuscript (Lines 47-48).

3. Line 106-107: Participants were asked not to perform exercise, such as running…”

How long (e.g. # of days prior and after the marathon running) were the participants not allowed to perform exercise? Were these participants trained or untrained people? If they were trained people, did the exercise prohibition of from this study induce any detraining or other effect(s)?

The participants were members of a college track and field club. They were prohibited from taking part in sports activity, training, or using any recovery practice such as massage for 8 days after the race to not induce any effect to the measurement. However, due to the fatigue after the full-marathon race, most participants in this study were not able to participate in sports activities during the observation period. Therefore, the exercise prohibition would be a recovery period from fatigue and it would not induce detraining effect.

4. Line 113: The baseline measurements were obtained 2 days before the race. Why 2 days prior specifically? Why not 1 day or any other time before the race?

The participants had registered for a full marathon race which took place far away from where they live; thus, they (examiner as well) had to move to the venue the day before the race. Therefore, the MR scanning was conducted 2 days before the race.

5. Line 160 and 162: Why two different post-hoc testings (Bonferroni and Dunnett)? Why not use only one of them and keep it consistent?

We used two different post-hoc tests depending on the different purpose of each comparison. Changes in T2 values and maximal knee flexion torque were compared by a three-way ANOVA (muscle × site × time point) and one-way ANOVA (4 time points), respectively. We performed Dunnett’s post hoc testing to compare the changes in these values from the baseline (PRE). On the other hand, absolute peak ΔT2 value for each site were compared by a one-way repeated measures ANOVA followed by Bonferroni post hoc testing because we intended to investigate the differences in each 11 sites.

6. Lines 218-226. The discussion of the non-significant intermuscular damage after marathon running needs to be expanded and more thorough. If this cannot be explained only by stretching magnitude, what else could be the mechanism/explanation?

We appreciate your suggestion that helps for improving our manuscript. We have expanded the discussion regarding this in accordance your suggestions as follow:

[Lines 215-230] Schache et al. [12] investigated the peak muscle–tendon stretch of the biarticular hamstring muscles at different running speeds and demonstrated that the peak stretch was greater for BFlh compared to ST and SM across a range of running speeds. They also found that there was no statistical difference in the magnitude of the maximum stretch between these muscles during slower speed below 18 km/h. The participants in the present study completed a full marathon at an average time of 4 h, 7 min, 55 s ± 47 min, 39 s (i.e., at an average speed of 11 km/h). Thus, the findings of Schache et al. [12] would support the result of the current study; however, the degree of muscle damage induced by prolonged running, such as after a full marathon, may not simply be explained based on the intermuscular difference in stretching magnitude: one possibility is the alternate muscle activity among the synergists due to the fatigue during prolonged running, while the previous study suggested the alterations in the neuromuscular activation patterns among the quadriceps muscles during the repetitive fatiguing exercise [20]. This phenomenon may occur in the hamstring muscles during marathon running. However, given the results of the mfMRI analysis indicating the intensity and/or pattern of exercise-induced muscle damage after exercise, whether the alternate muscle activity among the hamstring muscles occurs during the marathon race is uncertain. Even though it is a speculation, is worthy of further examination.

---

## [Decision Letter · Decision Letter 1]

11 May 2020

PONE-D-20-03671R1

Regional differences in hamstring muscle damage after a marathon

PLOS ONE

Dear Ms. Higashihara,

Thank you for submitting your manuscript to PLOS ONE. After careful consideration, we feel that it has merit but does not fully meet PLOS ONE’s publication criteria as it currently stands. Therefore, we invite you to submit a revised version of the manuscript that addresses the points raised during the review process.

I thank the authors for addressing the reviewers concerns on what is certainly a very interesting paper. It is clear the authors have worked hard to improve this manuscript. However, I have a few additional comments that need to be addressed as it relates to your statistical analyses/data interpretation.

I appreciate that you put the 95% CI on your figure but it still capturing the wrong variability. For example, in your peak T2 figure, you have the mean change and the 95% CI of the change (from Pre). That’s useful because that is the variability of the change as opposed to the variability of the sample at a given time point. So your other figures should begin at “0” and then show the change at each time point and the 95% CI of that change at each time point. So for strength…the variability would be reported as

Pre: 0

D1:-10 (-15.5, -4.5)

D3: -7.8 (-12.2, -3.4)

D8: -5.5 (-9.5, -1.5)

Related to the previous comment, in the manuscript you state that you performed a Dunnet’s post-hoc test…I am not sure that is typical of a repeated measures (within, within) design…are you sure you didn’t run a Bonferroni correction?

Related to the first comment, your effect size is calculated using what some would consider the wrong variability. For example, you are interested in stating the mean difference relative to the variability of that difference. Your calculation does not include that.  Based on how you are using the effect size, the calculation is fairly easy (change score divided by SD of the change score)…see below for strength as an example

Change to D1: 0.85

Change to D3: 0.83

Change to D8: 0.65

The values are pretty close to what you have but you are using the variability of the sample as opposed to the variability of the change…this can often really impact the effect size estimate (discussed here https://www.ncbi.nlm.nih.gov/pubmed/30358698).

In the abstract (line 29) you are saying that T2 significantly increased at the distal biceps femoris head within each individual muscle and have p values for each one. It would be more in line with your analyses if you just stated that the distal site swelled more compared to the middle or proximal sites because this did not differ across muscles...so in essence all of the muscles are pooled together at each site and compared.  This is also true in the results section. It would be clearer if you emphasized that this was a site x time effect (not a site x time x muscle) you were describing.

I thank the authors for addressing the 3-way interaction as this is directly related to their research question. The authors should be a bit clearer in the statistical analyses section for how they dealt with the proximal site for the BFsh…since there are no values for this site, how was this handed in the analysis?

We would appreciate receiving your revised manuscript by Jun 25 2020 11:59PM. To enhance the reproducibility of your results, we recommend that if applicable you deposit your laboratory protocols in protocols.io, where a protocol can be assigned its own identifier (DOI) such that it can be cited independently in the future. For instructions see: http://journals.plos.org/plosone/s/submission-guidelines#loc-laboratory-protocols

We look forward to receiving your revised manuscript.

Kind regards,

Jeremy P Loenneke

Academic Editor

PLOS ONE

Reviewers' comments:

Reviewer's Responses to Questions

**Comments to the Author**

1. If the authors have adequately addressed your comments raised in a previous round of review and you feel that this manuscript is now acceptable for publication, you may indicate that here to bypass the “Comments to the Author” section, enter your conflict of interest statement in the “Confidential to Editor” section, and submit your "Accept" recommendation.

Reviewer #2: All comments have been addressed

2. Is the manuscript technically sound, and do the data support the conclusions?

Reviewer #2: (No Response)

3. Has the statistical analysis been performed appropriately and rigorously? 

Reviewer #2: (No Response)

4. Have the authors made all data underlying the findings in their manuscript fully available?

Reviewer #2: (No Response)

5. Is the manuscript presented in an intelligible fashion and written in standard English?

Reviewer #2: (No Response)

6. Review Comments to the Author

Reviewer #2: (No Response)

7. PLOS authors have the option to publish the peer review history of their article (what does this mean?). If published, this will include your full peer review and any attached files.

Reviewer #2: No

---

## [Author Response · Author response to Decision Letter 1]

22 May 2020

Responses to Editor: 

We appreciate your insightful comments that helped improve the quality of our manuscript. We have conducted statistical reanalysis and revised the manuscript in accordance with your suggestions and hope that the revised manuscript is eligible for publication.

Our answers to the comments raised in the editorial report are as follows.

1) I appreciate that you put the 95% CI on your figure but it still capturing the wrong variability. For example, in your peak T2 figure, you have the mean change and the 95% CI of the change (from Pre). That’s useful because that is the variability of the change as opposed to the variability of the sample at a given time point. So your other figures should begin at “0” and then show the change at each time point and the 95% CI of that change at each time point. So for strength…the variability would be reported as

 Pre: 0

D1:-10 (-15.5, -4.5)

D3: -7.8 (-12.2, -3.4)

D8: -5.5 (-9.5, -1.5)

2) Related to the previous comment, in the manuscript you state that you performed a Dunnet’s post-hoc test…I am not sure that is typical of a repeated measures (within, within) design…are you sure you didn’t run a Bonferroni correction?

We appreciate your insightful suggestion. We apologize for our misunderstanding regarding the variability. We have revised the Figure 2 and Figure 4. In addition, after consideration, we have conducted the Bonferroni’s post-hoc tests in accordance with your suggestions and added the Table 2 that shows the T2 value for each site. The results were slightly altered (reverted to same result with previous version): the T2 values of the distal and middle sites increased significantly at D1 and D3 and recovered at D8, although those values of the proximal site remained constant. Although the results have been altered, we thought there is no need to change the directions of the discussion and conclusion from the previous version.

3) Related to the first comment, your effect size is calculated using what some would consider the wrong variability. For example, you are interested in stating the mean difference relative to the variability of that difference. Your calculation does not include that. Based on how you are using the effect size, the calculation is fairly easy (change score divided by SD of the change score) see below for strength as an example

Change to D1: 0.85

Change to D3: 0.83

Change to D8: 0.65

The values are pretty close to what you have but you are using the variability of the sample as opposed to the variability of the change…this can often really impact the effect size estimate (discussed here https://www.ncbi.nlm.nih.gov/pubmed/30358698).

Thank you for your insightful suggestion regarding the effect size. We have calculated the effect size as the mean change value from the PRE divided by the standard deviation of the change value and added these values in the Table 2 (for absolute T2 values). We also calculated the effect size using the same formula for the knee flexor torque in accordance with your suggestion. In addition, we referred the paper that you suggested (Dankel and Loenneke, 2018) in the revised manuscript.

4) In the abstract (line 29) you are saying that T2 significantly increased at the distal biceps femoris head within each individual muscle and have p values for each one. It would be more in line with your analyses if you just stated that the distal site swelled more compared to the middle or proximal sites because this did not differ across muscles...so in essence all of the muscles are pooled together at each site and compared. This is also true in the results section. It would be clearer if you emphasized that this was a site x time effect (not a site x time x muscle) you were describing.

Thank you for your comments. Because the results have been changed due to statistic reanalysis, we have revised the abstract and result section as follows:

[Line 29-32] 

Results indicated that no significant intermuscular differences in T2 changes were observed and that, regardless of muscle, the T2 values of the distal and middle sites increased significantly at D1 and D3 and recovered at D8, although those values of the proximal site remained constant. T2 significantly increased at distal and middle sites of the biceps femoris long head on D1 (p=0.030 and p=0.004, respectively) and D3 (p=0.007 and p=0.041, respectively), distal biceps femoris short head on D1 (p=0.036), distal semitendinosus on D1 (p=0.047) and D3 (p=0.010), middle semitendinosus on D1 (p=0.005), and distal and middle sites of the semimembranosus on D1 (p=0.008 and p=0.040, respectively) and D3 (p=0.002 and p=0.018, respectively).

[Lines 174-180]

Bonferroni’s post hoc analysis indicated that, regardless of muscle, the T2 values of the distal and middle sites increased significantly at D1 and D3 and recovered at D8 (no significant difference on D8 when compared with PRE), although those values of the proximal site remained constant. T2 significantly increased at distal and middle sites of the biceps femoris long head on D1 (p=0.030 and p=0.004, respectively) and D3 (p=0.007 and p=0.041, respectively), distal biceps femoris short head on D1 (p=0.036), distal semitendinosus on D1 (p=0.047) and D3 (p=0.010), middle semitendinosus on D1 (p=0.005), and distal and middle sites of the semimembranosus on D1 (p=0.008 and p=0.040, respectively) and D3 (p=0.002 and p=0.018, respectively). There was no significant difference in the T2 value of the middle BFsh muscle and the proximal site of the BFlh, ST, and SM at any time point compared with that of PRE.

5) I thank the authors for addressing the 3-way interaction as this is directly related to their research question. The authors should be a bit clearer in the statistical analyses section for how they dealt with the proximal site for the BFsh…since there are no values for this site, how was this handed in the analysis?

Thank you for your suggestion regarding the statistical analysis. We conducted the 3-way ANOVA excluding the proximal site of the BFsh by adjusting the degrees of freedom (df) (Site × muscle = 5). Table 1 in the manuscript shows the results of statistical analysis including df.

---

## [Editor Report · Decision Letter 2]

27 May 2020

Regional differences in hamstring muscle damage after a marathon

PONE-D-20-03671R2

Dear Dr. Higashihara,

We are pleased to inform you that your manuscript has been judged scientifically suitable for publication and will be formally accepted for publication once it complies with all outstanding technical requirements.

With kind regards,

Jeremy P Loenneke

Academic Editor

PLOS ONE
---

## [Editor Report · Acceptance letter]

15 Jun 2020

PONE-D-20-03671R2 

Regional differences in hamstring muscle damage after a marathon 

Dear Dr. Higashihara:

I'm pleased to inform you that your manuscript has been deemed suitable for publication in PLOS ONE. Congratulations! Your manuscript is now with our production department. 

Kind regards, 

on behalf of

Dr. Jeremy P Loenneke 

Academic Editor

PLOS ONE